# Looking into Endoplasmic Reticulum Stress: The Key to Drug-Resistance of Multiple Myeloma?

**DOI:** 10.3390/cancers14215340

**Published:** 2022-10-29

**Authors:** Guangqi Wang, Fengjuan Fan, Chunyan Sun, Yu Hu

**Affiliations:** 1Department of Hematology, Union Hospital, Tongji Medical College, Huazhong University of Science and Technology, Jiefang Avenue 1277, Wuhan 430022, China; 2Collaborative Innovation Center of Hematology, Huazhong University of Science and Technology, Wuhan 430074, China

**Keywords:** endoplasmic reticulum stress, unfolded protein response, drug resistance, multiple myeloma

## Abstract

**Simple Summary:**

Advances in treatment, especially with novel drugs, have dramatically improved the survival of multiple myeloma (MM) patients recently. However, frequent relapses and drug resistance remain unsolved issues. Endoplasmic reticulum stress (ERS) is elevated in MM compared to normal plasma cells, and is regarded as the Achilles’ heel of MM. This review summarizes the current knowledge of how ERS response influences the pathogenesis and drug-resistance of MM, and provides inspiration for novel therapeutic strategies to improve clinical outcomes of MM patients by targeting ERS.

**Abstract:**

Multiple myeloma (MM) is the second most common hematologic malignancy, resulting from the clonal proliferation of malignant plasma cells within the bone marrow. Despite significant advances that have been made with novel drugs over the past two decades, MM patients often develop therapy resistance, especially to bortezomib, the first-in-class proteasome inhibitor that was approved for treatment of MM. As highly secretory monoclonal protein-producing cells, MM cells are characterized by uploaded endoplasmic reticulum stress (ERS), and rely heavily on the ERS response for survival. Great efforts have been made to illustrate how MM cells adapt to therapeutic stresses through modulating the ERS response. In this review, we summarize current knowledge on the mechanisms by which ERS response pathways influence MM cell fate and response to treatment. Moreover, based on promising results obtained in preclinical studies, we discuss the prospect of applying ERS modulators to overcome drug resistance in MM.

## 1. Introduction

Multiple myeloma (MM) is the second most common hematopoietic malignancy, which occurs most often in elderly persons [1]. According to epidemiologic data from the International Agency for Research on Cancer, the current worldwide age-standardized incidence rates of MM vary from 0.54 to 5.3 per 100,000, which is associated with economic resources, patient education, and quality of health care [2]. Extensive efforts have focused on developing new drugs for MM during the past two decades. The emerging novel drugs, including proteasome inhibitors (PIs, i.e., bortezomib, carfilzomib, and ixazomib), immunomodulatory drugs (IMiDs, i.e., thalidomide, lenalidomide, and pomalidomide), and monoclonal antibodies against CD38 or SLAMF7, have improved the median overall survival time of MM, from 2.5 years before 1997, to 8–10 years currently [3,4]. However, MM remains incurable, due to relapses and resistance to therapies after different lines of treatment. According to the current guidelines for MM [5,6], drug combinations are used in first line. Treatment options for transplant-eligible patients include the three-drug regimen (triplet) VRd (lenalidomide + bortezomib + dexamethasone), or the monoclonal antibody against CD38 (daratumumab) plus triplet VTD (thalidomide + bortezomib + dexamethasone) or VRd. For transplant-ineligible patients, a doublet Rd (lenalidomide + dexamethasone) plus daratumumab or VRd can be used. The responses to treatment vary among MM patients, which can be classified as primary refractory (no response), partial response, and complete remission. For most patients who experience a period of improvement, the duration of response decreases with each line of therapy [7]. Patients who are clinically resistant to both bortezomib and first-line IMiDs have poorer outcomes [8]. These obstacles highlight the importance of exploring the mechanisms and tackling strategies of drug-resistance in MM.

MM cells are abnormal plasma cells that are characterized by an overproduction of monoclonal immunoglobulins. The malignant cells, as well as monoclonal proteins and cytokines secreted by the malignant cells, can lead to end-organ damage, including hypercalcaemia, renal dysfunction, anemia, and bone disease with lytic lesions (known as the CRAB criteria) [9]. Secretions of large amounts of immunoglobulins by MM cells exerts tremendous pressure on the endoplasmic reticulum (ER), an essential organelle that is responsible for accurately and durably maintaining a functional cellular proteome [10]. Any subtle disturbance to fine-tuning proteostasis may impact the fate of cells, and is prone to engender ER stress (ERS), which is viewed as the “Achilles heel” of MM [11]. Therefore, ERS response (ERSR), the integral and systemic cellular response to ERS, has become an attractive potential therapeutic target for MM [12]. To figure out how the ERSR helps resist cell death and promotes cell viability is essential for dealing with MM chemoresistance.

The unfolded protein response (UPR) that follows ERS, and ER-associated degradation (ERAD), are two quality-control machineries to maintain ER protein homeostasis whenever unfolded or misfolded proteins accumulate in the ER. During ERAD, misfolded proteins are recognized and degraded by the ubiquitin-proteasome system, or by lysosomes/vacuoles [13]. When ERAD fails to defend ER homeostasis, the UPR, a major transcriptional and signaling network, is activated [14]. Upon occurrence of the UPR, the cell cycle is arrested for cellular repair, protein translation is attenuated, and chaperones/co-chaperones are upregulated to assist protein folding [15]. In return, the UPR is able to activate ERAD and/or autophagy [15]. However, under the persistence of noxious factors and failure of self-adaptation, extensive and long-lasting ERS may induce general cellular responses, including cell cycle arrest, inflammation, and apoptosis [16]. Moreover, Ca^2+^ release from the ER, and its subsequent signaling cascades, are other important cellular events evoked under ERS, which also impact cell fate. In this review, we will discuss how MM cells evade drug-induced cell death to achieve tumor persistency through the ERSR, and the potential therapeutic strategy to overcome drug resistance in MM by modulating the ERSR. 

## 2. ERS and Drug-Resistance of MM

One essential function of the ER is the synthesis, folding, maturation, quality control, degradation, and proper delivery of secretory and transmembrane proteins [17]. Enormous biosynthetic load as well as multiple unfavorable factors (calcium imbalance, redox reaction, hypoxia, glycopenia, acidosis, etc.) place special demands on the ER, and promote the ERSR in order to restore homeostasis [18] (Figure 1). During the UPR, the principal part of the ERSR, chaperones are activated and dissociated from transmembrane sensor proteins on the ER membrane [18]. These changes lead to the activation of downstream signaling pathways, and to the reprogramming of a myriad of cell events, which may be pro-survival or pro-death [18]. Apart from the UPR, Ca^2+^ leakage from the ER is also activated under ERS, and is associated with tumor chemoresistance, which will be illustrated in “Section 4”.

MM cells are highly dependent on the UPR to alleviate ER stress. Compared to the normal population and patients with monoclonal gammopathy of undetermined significance (MGUS), the level of the UPR is significantly upregulated in cells of MM patients, and is essential for MM cell survival [19,20]. How drug-resistant MM cells modulate the UPR to evade cell death remains a research focus for novel strategies in MM therapy.

### 2.1. UPR Signaling: Canonical 3 Arms in MM

Classic UPR signaling pathways are initiated by three transmembrane sensors on the ER: (i) activating transcription factor 6 (ATF6), (ii) inositol-requiring kinase 1 (IRE1α), and (iii) PKR-like ER kinase (PERK) [21]. Under normal conditions, these sensors are bound by ER-resident chaperone glucose-regulated protein 78 (Grp78, also called immunoglobulin-binding protein (BiP)), member of the HSP70 (heat shock protein 70) family, to maintain an inactive state. When aggregation of misfolded proteins in the ER lumen occurs, UPR sensors detect the lesion and trigger the dissociation from Grp78. These alterations lead to the homo-dimerization and homo-oligomerization of IRE1α and PERK, respectively, and the migration of ATF6 to the Golgi apparatus, which induce the sophisticated downstream network of events [22]. The three pathways are artificially defined, and indeed a cross-interactive cascade modulating comprehensive cell responses to achieve adaption, or result in apoptosis (Figure 1).

The IRE1 pathway is highly conservative. Activated IRE1α exhibits its endoribonuclease nature, and excises a small intron from the x-box binding protein 1 (XBP1) mRNA, resulting in a template for the transcription factor-spliced XBP1 (sXBP1) [23]. sXBP1 is then transported into the nucleus, and orchestrates large transcriptional programs with hundreds of target genes that are involved in lipid biogenesis, chaperone synthesis, cell dormancy, ERAD, and apoptosis [24,25,26]. In addition, IRE1 may activate the IRE1-TRAF2 (TNF receptor associated factor 2)-JNK (c-Jun N-terminal kinase) cascade [27], thereby activating downstream pro-inflammatory transcriptional factors, as well as other mediators of autophagy or apoptosis. The activation of IRE1 and its downstream sXBP1 are essential for the ER expansion to ensure the normal secretory function of plasma cells during B-cell differentiation [28], and potentially contribute to the pathogenesis of MM [29]. Lower levels of XBP1 and sXBP1 in MGUS and MM plasma cells, compared to normal plasma cells, have been reported [30], indicating a possible situation where malignant plasma cells have adjusted to the pressure of aberrant synthesized paraproteins. Moreover, the IRE1-XBP1 axis has been proven to be a corresponding factor in MM bone disease [31]. The aforementioned evidence emphasizes the materiality and therapeutic capacity of the IRE1 pathway in MM.

When PERK is activated, it phosphorylates eukaryotic initiation factor 2 subunit α (eIF2α), a key initiator for protein translation [32]. The phosphorylation of eIF2α attenuates the global translation of most mRNAs to alleviate ER stress, except for activating transcriptional factor 4 (ATF4), which is preferentially increased [33]. ATF4 transcriptionally controls a wide range of adaptive genes that assist protein folding, regulate the metabolism of amino acid and glutathione, and increase cellular endurance to oxidative stress [34]. When cellular stress persists, ATF4 may lead to cell cycle arrest, senescence, or pro-apoptotic encoding [35,36]. ATF4 can activate C/EBP homologous protein/growth arrest/DNA damage-inducible protein 153 (CHOP/GADD153), a transcriptional factor that points to apoptosis [37]. In turn, GADD34, induced by ATF4-CHOP, enhances global protein translation by dephosphorylation of eIF2α, exacerbating protein load during ER stress and causing cell death [38]. Recently, another study identified QRICH1 as a newly found downstream factor of the PERK pathway, which dictates the outcome of ERS and cell fate by transcriptionally promoting UPR-mediated proteotoxicity [39]. 

ATF6 initiates the third pathway of UPR signaling [40]. When transported to the Golgi apparatus, activated ATF6 is processed and reveals its ATF6f domain, which then translocates to the nucleus and acts as a transcriptional factor to regulate the expression of other UPR genes [41,42]. Additionally, ATF6 can induce XBP1 transcription, which further initiates cell protection responses or apoptosis, constructing the relationship between the IRE1 pathway and the ATF6 pathway [43]. It is noteworthy that the three arms of the UPR are inter-modulated to impact cell fate.

Compared to the IRE1-XBP1 axis, the PERK-eIF2α-ATF4 branch and ATF6 are less studied in MM, and their accurate roles in MM pathogenesis have not yet been defined. Notably, the PERK pathway persists in the end-stage of UPR, and mediates UPR-triggered cell death, while the IRE1 and ATF6 pathways mainly act in the early phase [44]. Therefore, the PERK pathway appears to be a vital target for eliminating tumors. Moreover, investigations in B cells and plasma cells showed that B cells activate all three branches if the UPR is induced, whereas plasma cells specifically silence the PERK-dependent pathway via differentiation-induced signals [45]. Interestingly, although MM cells have common phenotypic characteristics with plasma cells, PERK is highly expressed in MM cells [46]. Taken together, altered conditions of the UPR in MM indicate that the UPR participates in MM tumorigenesis, and is a vulnerable target for therapeutic strategies. 

### 2.2. Altered ERSR Activation in Drug-Resistant MM

Due to the nature of MM cells, most backbone treatment strategies of MM are based on proteasome inhibitors (PIs), immunomodulators (IMiDs), and corticosteroid dexamethasone (Dex), which directly or indirectly target the ubiquitin-proteasome system. Bortezomib (BZ) is one of the first-in-class PIs approved by the FDA in 2003. It functions by selectively and reversibly inhibiting the 26S proteasome, disrupting the ERAD process and protein homeostasis of MM, thus promoting apoptosis [47]. In addition to immune modulation, the effects of IMiDs include direct anti-MM activity and disruption of MM-bone marrow microenvironment interactions. An important mechanism of action is that IMiDs bind to CRBN-DDB1 E3 ubiquitin ligase and stabilize CRBN by inhibiting its own ubiquitination, thus increasing the degradation of key transcription factors IKZF1 (Ikaros) and IKZF3 (Aiolos) in MM through the ubiquitination-proteasome system [48]. Importantly, IMiDs increase H_2_O_2_ levels in MM cells expressing CRBN, leading to immunoglobulin dimerization and accumulation, followed by ERS induction, which then triggers apoptosis through the BH3-only protein Bim [49]. Moreover, in MM cells, lenalidomide treatment significantly elevates the expression of caspase-4, which has been implicated in ERS-induced apoptosis [50]. Dex has been found to trigger upregulation of proteins that are involved in protein folding and trafficking, as well as apoptosis in MM cells [51]. Combined with BZ, IMiDs, Dex, or other agents are able to induce multiple alterations in MM cells, including ERS, ROS (reactive oxygen species) accumulation, calcium leakage, and cytochrome c release [52]. The combination of BZ, lenalidomide and Dex (VRd) has been the first-line chemotherapy for newly diagnosed MM, but has shown less effectiveness in RRMM. Therefore, many efforts have been made to explore alterations of drug-resistant MM cells, and to innovate strategies to overcome multidrug resistance.

Dense variations of the UPR have been detected in drug-resistant MM cell lines and in RRMM patients. MM cells that produce more immunoglobulins respond better to PIs, both in vitro and in clinical observations [53,54]. PI-resistant MM cells display less dependence on the ERSR to restore homeostasis, compared to PI-sensitive MM cells. Multiple studies have proved that the sXBP1 level is downregulated in BZ-resistant cell lines and BZ-refractory MM patients [55]. Moreover, a low XBP1 spliced/unspliced ratio (sXBP1/uXBP1) has been associated with longer OS and better clinical outcomes in MM patients treated with thalidomide [56], making it a surrogate biomarker for predicting patient response to treatment. Jonathan et al. revealed that knockdown of IRE1α strongly increases sensitivity to BZ and lenalidomide in MM cells [57]. Tang et al. have shown that the inhibition of IRE1-XBP1 significantly impairs viability, and overcomes the resistance to PIs in MM cell lines [58]. These data suggest the potential of IRE1α inhibition to improve the efficacy of MM treatment with drugs that disrupt protein homeostasis, such as BZ and IMiDs. Moreover, a genomic analysis of treatment-refractory MM patients reported two inactivating mutants in XBP1, P326R, and L167I, which were a transactivation domain mutation and a splicing site mutation, respectively, both impairing XBP1 [59]. Nevertheless, the direct manipulation of XBP1 and the silencing of either PERK or ATF6 did not show active reversible effects of BZ resistance in collective experiments [60]; further investigation is required. 

A transcriptome analysis revealed that the expression levels of ATF3 and ATF4 were lower in a short-PFS group (< 6 months) compared to a long-PFS group in MM patients. In drug-sensitive MM, BZ can transactivate ATF genes, and pose overwhelming ER stress to cause cell death [61]. However, in drug-resistant MM, BZ treatment may suppress ATF3 and ATF4 in an RNA-interfering manner [21,62]. Together, these findings indicated the specific mapping of the UPR in MM chemotherapeutic cytotoxicity.

Furthermore, active proteasomes, the central players of ERAD, are frequently mutant in resistant MM cells, which may directly undermine the efficiency of PIs. A point mutation in gene encoding PSMB5 (proteasome b5 subunit) poses a conformational change within domain-mediating proteasome-BZ binding, and is associated with the PI-resistant phenotype in MM cell lines [63]. Downregulating PSMB6 or knocking out PSMB5 was able to reverse PSMB5 mutation-induced BZ resistance [64], providing potential novel strategies to overcome resistance to PIs in MM. Evidence has also demonstrated that silence of PSMB5 promotes the activation of M1 macrophages in vitro, which indicates the dual role of PSMB5, and its potential impact on cancer immunotherapy [65]. From this perspective, it would be interesting to investigate qualitative alterations of proteasome subunits in drug-resistant MM cells.

Altered chaperones/co-chaperones also impact drug sensitivity in MM. Increased Grp78 levels and enhanced Grp78-mediated autophagy after BZ treatment have been revealed in MM cell lines and biopsies, which is a pro-survival adaptation [66]. Cell division cycle 37 (Cdc 37), a co-chaperone of HSP90, is less expressed in BZ-resistant clinical samples [67], which may be a result of clonal selection, and confers BZ resistance through downstream effector XBP1s [68]. In addition, 70 kDa HSP (HSP70, HSP72, or HSPA1) prolongs XBP1 splicing and helps MM cells evade ERS-induced apoptosis [69]. It has been well established that both cellular and non-cellular components of bone marrow microenvironments promote the development of drug resistance of MM [70,71]. Adherence of MM cells to bone marrow stromal cells (BMSCs) or fibronectin has been implicated in BZ, as well as melphalan resistance, via integrin-dependent upregulation of HSP70 [72]; this indicates a role of ERSR signaling in cell adhesion-mediated drug resistance in MM. A recent study illustrated a possible upstream mechanism of altered chaperones in BZ-resistant MM—S-glutathionylation of Grp78/BiP [73]—which could inspire ideas about protein/chaperone modification in MM drug sensitivity.

## 3. MM Cell Survival and ERSR

The most common mechanism of chemotherapies is to induce apoptosis of MM cells. Based on the initial stage, there are three main pathways that mediate apoptosis: (i) the mitochondrial pathway, (ii) the endoplasmic reticulum pathway, and (iii) the death receptor pathway; on the basis of the original signaling region, there are extrinsic (iii) and intrinsic (i, ii) pathways [14]. In general, apoptosis signaling pathways converge at caspases, a family of cysteine aspases [74]. A better understanding about how tumor cells avoid apoptosis may facilitate novel strategies to resolve chemoresistance. 

The ERSR dynamically determines cell death and survival by regulating autophagy or apoptosis, and plays a role in chemotherapy-tolerant MM. In this part, we will summarize how MM cells survive the ERSR to develop drug resistance.

### 3.1. ERS-Induced Apoptosis and MM Drug Resistance

When the severity of ERS is beyond the tolerable threshold, and when this stress persists, the ERSR will activate pro-apoptotic pathways. UPR-mediated apoptosis is the main part of ERSR-induced cell death. The UPR may induce cell apoptosis via three ways: (i) transcriptional activation of the CHOP/GADD153 gene [75]; (ii) the IRE1-TRAF2-JNK pathway [76]; and (iii) ER-dependent caspase-12 (or caspase-4 in human cells) mobilization [77]. 

Drug-tolerant MM cells are able to reprogram the ERSR signaling cascades and evade apoptosis, achieving tumor persistency. Therefore, by targeting ERSR signaling, the cell fate of drug-resistant MM cells may switch from pro-survival to pro-death, thus mitigating the chemotherapy tolerance of MM. When treated with PIs, MM cells go through ERS-mediated apoptosis. Upregulation and activation of CHOP/GADD153 and JNK have been detected in apoptotic MM cells that were treated with PIs, which is frequently associated with NF-κB modulation [78,79]; this makes it a therapeutic target for drug resistance. Several studies showed that activation of the Jun-JNK pathway by adaphostin contributed to caspase-dependent apoptosis in MM cell lines [80], and that by activating JNK, drug tolerance in MM was alleviated [81]. Study aboutcaspase-12 or its homologous protein caspase-4 in MM isrelatively sparse. In MM cell lines, it was reported that ERS-induced apoptosis did not require caspase-12 or caspase-4 [82]. Nevertheless, as caspase-12, caspase-9, and caspase-3 have been verified to have superior antitumor efficiency, and demonstrated roles in drug resistance in a variety of cancers, they should not be neglected, and are possible candidates for combined therapy in MM to enhance drug sensitivity [83,84]. Furthermore, as a potent downstream functional factor of the ERSR, the BCL-2 (B-cell lymphoma 2) family plays a dynamic role in regulating cell death and MM drug resistance. Elevations in BCL-2, BCL-xL, and MCL-1 proteins have been found in resistant MM cells [85], with BCL-2 being the major factor that mediated Dex resistance [86]. Dex is able to sensitize MM cells to venetoclax, a BCL-2 inhibitor, by promoting BCL-2 cell dependence in an altered Bim-binding pattern [87]. Therefore, targeting the terminal reaction cascade of cell death is a rational strategy to ameliorate therapeutic efficiency.

### 3.2. ERS-Mediated Autophagy and Survival of Drug-Resistant MM Cells

During the ERSR, if ERAD and the UPR fail, autophagy is considered the last attempt to restore ER homeostasis. In the case of sustained ERS, autophagy is activated, and aberrant ER will be partially engulfed into autophagosomes and then transported to lysosomes for degradation. The degraded fragments will be reutilized for newly assembled ER, promoting normal ER status [88]. UPR-mediated autophagy has prominently contributed to tumor persistence, due to its pro-survival effect [89]. It has been illustrated that activation of autophagy is a significant mechanism of MM cell survival and drug resistance [90]. The simultaneous inhibition of the proteasome system by BZ and autophagy by hydroxychloroquine (HCQ) revealed synergistic and superior cytotoxicity of the two agents [91], providing preclinical evidence for the application of HCQ in RRMM. Consistent results also revealed that promoting autophagy may lead to aggravated Dex resistance in MM [92]. Therefore, having a thorough understanding about how autophagy is activated under ERS and contributes to tumor persistence is crucial for exploiting novel resistance-reversing strategies.

All three pathways of the UPR are engaged in the initiation of autophagy, and ATF6 contributes the least compared with the other two UPR arms [93]. In a retrospective cohort study of 89 MM patients, the autophagic markers Beclin-1 and LC3, that were detected from biopsies, have been identified as favorable prognostic predictors [94]. Suppressed PERK-eIF2α-autophagy axis by Toll-like receptor-4 causes enhanced cell survival and compromised BZ efficiency in MM [95]. Blocking the translation of ATF4 by myxoma virus infection shows promising antitumor efficiency in PI-resistant MM [96]. Moreover, PIs have been proven to elevate SREBP1/2, which is induced by ATF4 and converges via the mechanism of action of lipid-modulating drugs [97]. It is noteworthy that during long-term ERS, activated ATF4 may also activate CHOP, and lead to cell death in MM. Suppression of the PERK-ATF4 axis may contribute to prolonged cell survival and aggravated BZ resistance [95]. 

The IRE1 pathway is as important as PERK is, in regulating autophagy. In MM cells, inhibition of IRE1-XBP1 could lead to attenuated PERK-dependent autophagy and promoted cell death [98], indicating a cross-link between the IRE1 and PERK pathways. Some researchers considered IRE1-TRAF2-JNK as an indispensable pathway [99], with markedly decreased autophagosomes in JNK-inhibited cells [100]. Similar to XBP1, JNK activates autophagy by mobilizing BCL-2 and Beclin 1, highlighting the therapeutic potential of targeting the two molecules. By inducing Beclin 1-mediated autophagy, Profilin-1 is able to induce drug resistance in MM [101]. Besides, evidence has shown that inhibition of autophagy by plitidepsin is associated with the downregulation of multiple UPR cascade proteins in MM cell lines [102]. 

Moreover, in MM cells, chaperone-mediated autophagy, a subtype of autophagy that is specific to degradation of cytosolic proteins initiated under ERS, has been demonstrated to confer resistance to BZ [103]. However, a couple of studies have shown that ERS-mediated autophagy may lead to MM cell death as well. For example, Fu et al. reported that enhancing ERS-promoted autophagy in MM could compromise cell proliferation and ameliorate drug resistance [104]. Michallet et al. illustrated that extinction of the UPR by sensor knockdown could induce a form of non-apoptotic cell death that is executed by autophagy in MM, which is associated with an intrinsic apoptotic pathway and mitochondria [105]. This evidence suggests the bidirectional function and perplexing role of autophagy. Therefore, further investigation is warranted, in order to elucidate the molecular mechanisms underlying ERS-mediated autophagy, and to determine how MM drug resistance can be overcome by modulating the related signaling.

### 3.3. Survival/Apoptosis Balance under ERS and Its Implication for MM

In general, under ERS, cells manage to survive by Beclin 1-mediated autophagy, or undergoing a suicide process that is executed by caspases. There is a sophisticated and robust manipulating mechanism which decides between autophagy/survival and apoptosis. In order to cure tumors and alleviate chemoresistance, it is rational to switch cell fate from pro-autophagy to pro-apoptosis. Thus, a better understanding about the factors that impact the survival/apoptosis switch is of great importance. 

Several ER-associated factors have been linked to autophagy/apoptosis switch, involving interplay among Beclin 1, BCL-2, and caspases, the most related factors to autophagy and apoptosis. The Beclin 1/BCL-2 complex has been frequently defined as a pro-survival factor. For example, knocking down Beclin 1 will reverse autophagy-mediated drug resistance in MM [106]. Hence it is also a rational approach to attenuate autophagy by tackling the Beclin 1/BCL-2 complex, in order to enhance drug efficiency. A hierarchical study evaluated the sensitivity of MM cells to inhibitors of anti-apoptotic proteins in the BCL-2 family (BCL-2, BCL-XL, BCL-W, A1, and MCL-1), and showed that the MCL-1 inhibitor is the most efficient among them [85], encouraging further study into inhibiting pro-autophagic proteins. Repression of MCL-1 in MM cells is associated with protein translation inhibition, and is coupled with tumor sensitivity to the ER stressor thapsigargin, which is able to induce mitochondrial apoptosis through MCL-1/Bak interaction [107]. Promisingly, several BCL-2 and MCL-1 inhibitors are currently being investigated in clinical trials (See Section 5). Moreover, a mathematical model of the BCL2-Beclin 1-caspases network has been assembled to qualitatively predict the behavior of the dynamic system. The discontinuous switch from one stable cellular state to another revealed that autophagy and apoptosis cannot coexist under any level of cellular stress [108]. This special interdisciplinary study about autophagy/apoptosis switch provides a novel strategy to monitor the cell fate of MM toward apoptosis in a quantitative approach. By studying the precise molecular alterations between different ERSR phases, there may be further insights regarding MM tumor persistency.

It is noteworthy that one of the possible downstream events of the ERSR is cell cycle arrest. Indeed, cell dormancy is another factor that impacts drug efficiency. Upon pressure of cytotoxic drugs, especially genotoxic agents, cell cycle checkpoints will examine the cue and decide whether to pause cellular proliferation, in order to restore homeostasis. In cancer cells, this kind of durable proliferation arrest or reversible “death” promotes post-therapy tumor repopulation, and is one of the main mechanisms that contribute to drug resistance [109]. In MM, misbehaving cells in active sleep refer to the multiple myeloma stem cell-like cells, which are believed to be the major cause of minimal residual disease [110]. MM cells of immature stem-like phenotypes tend to be more quiescent, which is linked to clinical drug resistance [111]. Although tumor stem cells have been considered to be the core of tumor regrowth, how chemotherapy impacts the tumor stemness is unclear. The role of heterogeneity and cell cycle modulation in various cancers has been increasingly emphasized; however, the drug-tolerant persistence has not yet been well-illustrated in MM cells [112]. To be noticed, it is also common to define growth arrest as a negative factor for tumor proliferation in MM [113], suggesting the dual role of cell dormancy. On one hand, tumor cells remain stable, and disease becomes less progressive; on the other hand, dormant cancer cells are more persistent than active ones. The elusive role of cell cycle arrest in MM drug resistance requires further investigation.

## 4. ERS-Triggered Ca^2+^ Leakage and Its Role in Modulating MM Cell Fate

ER is the largest intracellular reservoir of ionic calcium (Ca^2+^), and calcium imbalance is one of the major cellular events of ERS. In normal conditions, the ER monitors calcium flow mainly via transporter–based mechanisms, including activated metabolic receptor-mediated Ca^2+^ release (mainly through inositol-1,4,5-trisphosphate IP3 receptor, IP3R) and Ca^2+^-induced Ca^2+^ release (through ryanodine receptor, RyR) [114,115]. The preservation of Ca^2+^ balance involves multiple buffers and sensors, including ER-associated proteins such as BiP [116]. Below, we discuss how ERS-triggered Ca^2+^ leakage impacts cell fate, particularly MM cells, and the possible approaches to modulate MM drug sensitivity by targeting Ca^2+^ signaling (Figure 1). 

In unfavored conditions, such as insufficient nutrients/energy, calcium leakage from the ER is potentially one of the downstream events that is mediated mainly by IP3R or RyR [117]. Then, the released Ca^2+^ is transmitted to local mitochondria through the mitochondria-associated ER membranes [118], or further induces Ca^2+^ release in a positive feedback manner, which is associated with cytochrome c [119]. Subsequently, Ca^2+^ uptake occurs through activated mitochondrial Ca^2+^ uniporters (MCUs), and the overloaded calcium within mitochondria leads to sustained opening of permeability transition pores (PTP) and rupture of the outer mitochondrial membrane [120]. In the end, the pro-apoptotic proteins are released from mitochondria into the cytosol, and initiate a resultant cascading mitochondrial apoptotic pathway, i.e., formation of cytochrome c-Apaf1-caspase-9 complex, which initiates proteolytic events [121]. Alternatively, Ca^2+^ released from the ER in the cytoplasm may directly activate calpain, which then activates caspase-12 and induces apoptosis [118]. Within the sophisticated network, the BCL-2 family plays a dual role in modulating apoptosis, with anti-apoptotic BCL-2 protein compromising Ca^2+^ release, as well aspro-apoptotic BAX and BAK promoting mitochondrial Ca^2+^ uptake and BAD sensitizing mitochondrial PTP to Ca^2+^ [122,123,124]. 

Ca^2+^ signaling may also interfere with autophagy. Ca^2+^ modulates autophagy through a perplexing signaling network that has a bidirectional effect, depending on the cell state. Generally, spontaneous Ca^2+^ may reduce the level of autophagy through the mitochondrial pathway under normal conditions. During ERS, Ca^2+^ is released into the cytosol, and the incrementing calcium concentration activates cytoplasm-located death-associated kinase 1 (DAPK1), a calcium/calmodulin-dependent serine/threonine kinase that enhances autophagy [125,126]. As an essential ER membrane calcium channel, IP3R has been reported to have both positive and negative regulatory effects on autophagy. Most prominently, IP3R activation that is followed by an uploaded cytosolic Ca^2+^ concentration mobilizes the calmodulin-dependent kinase, kinase β (CaMKK2), which arouses a CaMKK2-AMPK-mTOR cascade, and consequently, autophagy [127]. Alternately, IP3R may also enhance the formation of BCL-2-Beclin 1 complex, which is anti-autophagic [128]. Apart from IP3R, calreticulin, a heat-shock protein or chaperone that is mainly located on the ER, is significantly upregulated under ERS, promoting the formation of autophagosomes and autophagy flux [129]. 

Ca^2+^ channels and transporters have been implicated in MM cell proliferation, dissemination, drug sensitivity, and clinical outcomes [130]. Activated MCUs can abate BZ resistance and promote cell death in MM [131,132]. In addition, Orail1, a store-operated Ca^2+^ entry channel that is located on the ER and the plasma membrane, regulates the motility and metastasis of MM; meanwhile, silencing Orail1 leads to cell cycle arrest, and to apoptosis of MM cell lines [133,134,135]. Even though there is no direct evidence that Dex impacts calcium transport in MM, it has been reported that higher concentrations of Dex increase the cytoplasmic Ca^2+^ in MM cell lines [136]. Moreover, activation of DAPK1, a downstream protein of Ca^2+^ leakage, leads to tumor suppression of MM [137]. A recently reported promising approach to reverse tumor drug resistance involves artificially operating intracellular calcium distribution, for example by “calcium ion nanogenerator”, a versatile calcium bursting method, which may be a potential strategy for MM treatment [138]. 

## 5. Emerging Strategies to Overcome MM Drug Resistance by Targeting ERS

Quickly increasing preclinical evidence can optimize the development of novel therapeutic strategies. The current strategy to overcome drug resistance in MM by targeting ERS is the combination of an ERS-modulating drug, including heat-shock protein/chaperone inhibitors, deubiquitinating enzymes (DUBs), ubiquitin activating enzyme (UAE) inhibitors and autophagy inhibitors, with common first-line PIs, Dex, or IMiDs in clinical trials or practice (Table 1).

The anti-myeloma effects of HSP90 inhibitors support their use in clinical trials. IPI-504 [139], PU-H71 [140], and SNX-2112 [141] show significant cytotoxicity in both general and drug-resistant MM cell lines, while AUY922, 17-AAG, and KW-2478 exhibit positive effects in clinical trials. 17-AAG/tanespimycin, a small molecule inhibitor that acts by inhibiting the chaperoning function of HSP90 which helps restore homeostasis, is the first HSP90 inhibitor to be evaluated in clinical trials. In combination with BZ, 17-AAG showed anti-myeloma effects in a phase I/II study of patients with refractory MM, and was well-tolerated [142]. AUY922 was involved in a phase I/Ib study completed in 2011, combined with BZ and Dex. However, AUY922 showed no complete response in MM patients, and its combination with the recommended dose of BZ was not tolerated [143]. KW-2478 in combination with BZ exhibited good tolerance, and an overall response rate of 39.2% (complete remission rate 3.8%), which was a modest result [144]. Apart from HSP90 inhibitors, an HSP70 inhibitor called MAL3-101 demonstrated anti-myeloma effects both in vitro and in vivo, combined with a PI [145], indicating preclinical evidence for HSP70 as therapeutic target. 

Grp78/Bip monoclonal antibodies, such as PAT-SM6, have been tested in RRMM combination regimens. The immunohistochemistry of clinical samples showed that the Grp78 level increases with disease progression, and is strongly elevated in patients with RRMM and extramedullary involvement [146]. A dose-escalating phase I trial of single-agent PAT-SM6 showed promising efficiency in disease stabilization [147]. Based on the verified anti-MM effect of PAT-SM6, a further preclinical study illustrated the superior efficiency of combining Dex, PAT-SM6, and lenalidomide in RRMM, which was proven in a real case of late-stage MM with extramedullary involvement [146]. Another emerging target is Grp94/Grp96, which has already been illustrated as a molecular hallmark of MM [148]. With preclinical evidence that demonstrated the effects of inhibiting Grp94 in MM cell lines [149], Grp94 inhibitors are highly expected to be involved in future clinical trials. 

IRE1α inhibitors exhibit promising antitumor efficacy, and augment the response of MM to established backbone regimens. The IRE1α endoribonuclease domain inhibitor MKC3946 has shown promising effects in restraining MM cells, without toxicity to normal mononuclear cells, while enhancing the cytotoxicity triggered by BZ or 17-AAG [150]. Furthermore, compound 18, an IRE1α kinase inhibitor, attenuates tumor growth and sensitizes MM to BZ and Len [57], demonstrating optimistic clinical and translational prospects.

DUBs are able to reduce the ubiquitinated protein load in MM cells, which may contribute to BZ resistance. Therefore, targeting DUBs is an attractive anti-MM strategy. B-AP15, a novel 19S subunit inhibitor, is able to generate a compromising effect on tumor viability, and sensitizes BZ-resistant MM cells to BZ therapy [151]. Similar effects have been observed with copper pyrithione, an inhibitor that targets both 19S proteasome and 20S proteolytic peptidase [152]. The DUB inhibitor P5091, which targets USP7, alternatively interferes with ubiquitin binding, and has been confirmed to have a BZ resistance-reversing effect in vitro and in vivo [153]. Another approach to aggravate protein load by modulating the ubiquitin-proteasome system, is to inhibit the UAE. TAK-243, a novel inhibitor for UAE, overcomes drug resistance, and shows activity against in vitro and in vivo models of MM [154], supporting its translation to clinical application.

Autophagy inhibitors are a large group of molecules that involve various mechanisms to downregulate pro-autophagic signals, or directly block autophagy. Despite undefined mechanisms, hydroxychloroquine (HCQ)/chloroquine (CQ) that is derived from the heterocyclic aromatic compound quinoline, may alkalinize lysosomes and disrupt the autophagic proteolysis in MM by inhibiting the formation of autophagosomes [155]. Promisingly, HCQ/CQ showed synergistic antitumor effects with rapamycin and cyclophosphamide in a subsequent minor clinical trial [156]. Moreover, in a phase I study, HCQ/CQ and BZ combined regimen exhibited enhanced antitumor efficiency in patients with RRMM, and compromised autophagosome formation in clinical samples, which was consistent with preclinical evidence [157]. 

Inhibitors that target anti-apoptotic proteins, especially BCL-2, are also clinically adapted anti-myeloma agents. Venetoclax is a small molecule of pyrrolopyridine that mimics native ligands of BCL-2, and binds to it with high selectivity, thereby repressing BCL-2 activity and restoring tumor apoptotic processes. For heavily pretreated MM patients, especially those harboring t(11;14), a phase II trial has illustrated promising results for venetoclax [158]. Furthermore, there were several other BCL-2 inhibitors that were assessed in clinical trials, with programs on S55746 and anti-apoptotic protein derivatives already completed; however, none of them displayed positive results. S55746 has been investigated in a phase I trial in patients with CLL, B-cell non-Hodgkin lymphoma or MM, but no results were posted. The phase I trial of a therapeutic peptide vaccination derived from anti-apoptotic proteins in patients with relapsed MM yielded immune responses with modest antitumor effects [159]. In addition, elevated MCL-1 is significantly associated with poor prognosis and drug resistance in MM patients [160]. For first generation MCL-1 inhibitors, it is a challenge to cope with drug specificity and off-target events. Recently, there were several MCL-1 inhibitors being tested in clinical trials, including AMG-397 [161], ABBV-467 [161], and MIK-665 [161]. However, trials with AMG-397 and ABBV-467 were terminated, and results of MIK-665 are unavailable. 

Chimeric antigen receptor (CAR) T-cell therapy has emerged as a novel option for patients with RRMM. CAR-T cells redirected to cell surface HSP70 and Grp78, which are induced by ERS and frequently expressed on various malignant and aggressive cells, were validated to have promising antitumor effects in solid tumors [162] and acute myeloid leukemia [163]. Although there is still lack of evidence in MM, it has been validated that cell surface Grp78 is significantly upregulated in plasma cells of patients with MM, compared to those with MGUS. Furthermore, a recent study has shown that ablation of PERK promotes antitumor T-cell responses by inducing paraptosis and type I interferon, suggesting that antitumor immunity could be modulated by the ERSR [164]. Inhibition of the IRE1-XBP1 pathway effectively repolarizes M2-tumor associated macrophages, and elevates the antitumor efficacy of PD-1 antibodies [165], highlighting the favorable dual impact on both tumoral and intratumoral immune cells by targeting the ERSR. Collectively, increasing evidence has implicated the potential of targeting ERSR-associated proteins to improve the efficacy of cancer immunotherapy. 

Unfortunately, there is no current evidence about applying Ca^2+^-targeted modulators to MM. Apart from directly manipulating ERSR-related candidates, HIV protease inhibitor nelfinavir has been found to enhance the UPR, and to sensitize MM cells to PIs [166]. A phase I trial that combined nelfinavir with BZ in treating patients with RRMM has been carried out; results showed promising antitumor activity [167]. Preclinical evidence and clinical trials together verified that ERS-modulating agents have great potential in aggravating cellular stress, and alleviating drug resistance in MM. It should be noted that ERS targets are frequently multi-functional, and bear a wide range of biological effects. Thus, targeting ERS candidates may bring about unexpected side effects.

**Table 1 cancers-14-05340-t001:** Agents targeting ERS to alleviate MM drug resistance. Agents alleviate drug resistance in MM by modulating ERS-associated proteins. Their target proteins and research information, including recent research phase and application of combined drugs, are listed. BZ, bortezomib; Dex, dexamethasone; Len, lenalidomide.

Agent Name	Drug Target	Research Status in MM		Reference
		Research Phase	Combination Drug	
IPI-504	HSP90	Preclinical phase	Single	[139]
PU-H71	HSP90	Preclinical phase	Single	[140]
SNX-2112	HSP90	Preclinical phase	Single	[141]
17-AAG	HSP90	Clinical trial phase I/II	BZ	[142]
AUY922	HSP90	Clinical trial phase I/Ib	Dex + BZ	[143]
KW-2478	HSP90	Clinical trial phase I/II	BZ	[144]
MAL3-101	HSP70	Preclinical phase	BZ	[145]
PAT-SM6	Grp78	Clinical trial phase I	Single	[146]
Radamide analogs	Grp94	Preclinical phase	Single	[149]
MKC3946	IRE1α	Preclinical phase	BZ	[150]
Compound 18	IRE1α	Preclinical phase	BZ or Len	[57]
B-AP15	DUBs (19S)	Preclinical phase	BZ	[151]
Pyrithione	DUBs (19S, 20S)	Preclinical phase	Single	[152]
P5091	DUBs (USP7)	Preclinical phase	BZ	[153]
TAK-243	UAE	Preclinical phase	Single	[154]
HCQ/CQ	Autophagy	Clinical trial phase I	BZ	[157]
Venetoclax	BCL-2	Clinical trial phase I/II	Dex	[158]
Peptide vaccination	Anti-apoptotic proteins	Clinical trial phase I	BZ	[159]
AMG 397	MCL-1	Clinical trial terminated	Dex	[161]
ABBV-467	MCL-1	Clinical trial terminated	Single	[161]
MIK-665	MCL-1	Clinical trial phase I	Single	[161]
Nelfinavir	HIV protease	Clinical trial phase I	Single	[167]

## 6. Conclusions and Perspectives

As a highly secretory tumor, MM is able to adapt to unfavorable conditions through modulating ERSR signaling. MM cells frequently develop drug resistance and therapy tolerance through inherent mechanisms, including ERAD, attenuated UPR, and ERS-induced autophagy, as well as ER-leaked Ca^2+^ signaling. Widely used anti-MM therapies showed effects in inducing ERSR-mediated apoptosis, though they were not originally designed for that. Thus, targeting ERSR-related molecules is a rational approach for overcoming drug resistance. Promising preclinical studies are paving the way for ERSR-targeted novel agents, whereas the results exhibited by clinical trials are middling. Despite being carefully designed and monitored, the selectivity and toxicity of novel drugs still pose a major challenge when combined with typical anti-MM regimens, and this warrants further evaluation. With respect to the heterogeneity of tumors, tailoring larger-scaled clinical trials to subdivided drug-resistant MM patients is potentially practical in future studies. High-throughput gene and protein profiling may identify gene-expression signatures that characterize the biology of RRMM in individual patients, which is vital for personalized targeted therapies, and helps improve clinical responses to combination therapy with ERS modulators. Notably, apart from directly triggering tumor cell death, manipulating ERS may also induce enhanced immune responses in tumor microenvironments, encouraging further investigation into ERS and immunotherapy in MM.

## Figures and Tables

**Figure 1 cancers-14-05340-f001:**
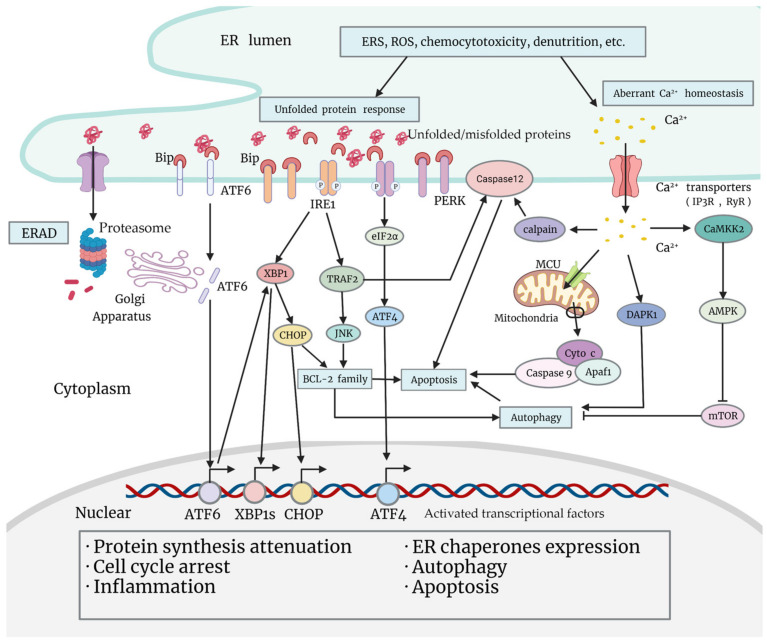
Classical signaling pathways of the unfolded protein response, and Ca^2+^ leakage under ERS (endoplasmic reticulum stress). Various adverse factors cause ERS within tumor cells, such as chemocytotoxicity, nutrition deprivation, production of ROS (reactive oxygen species), etc. Increasing ERS leads to ERAD (ER-associated degradation) and arousal of the ERSR. The UPR is the main part of the ERSR. When unfolded or misfolded proteins aggregate in the ER lumen, Grp78/BiP dissociates from sensor proteins ATF6, IRE1, and PERK on the ER membrane, triggering a downstream signaling cascade. Activated ATF6 migrates to the Golgi apparatus, and is cleaved to transform into a transcriptional factor, which regulates the transcription of UPR genes including XBP1. Activated PERK is able to phosphorylate eIF2α and mobilize ATF4 to promote cellular adaptation. Functional transcriptional factors that are mobilized in the UPR will translocate to the nucleus, and exert multiple biological effects that promote protein synthesis attenuation, ER chaperones expression, cell cycle arrest, inflammation, autophagy, or apoptosis. Apart from activating XBP1 and the subsequent expression of CHOP, IRE1 is also able to promote pro-inflammatory JNK activation through activating TRAF2, which may impact cell fate by regulating BCL-2 family proteins or caspase-12 (or caspase-4 in human cells). Under ERS, another important alteration that impacts cell fate is Ca^2+^ leakage from the ER. Ca^2+^ leaked from the ER may induce apoptosis by activating the mitochondria pathway or caspase-12 (or caspase-4) on the ER membrane, or promote autophagy by activating DAPK1 or by inhibiting mTOR through the CAMKK2-AMPK pathway.

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
