# Peer review of "Looking into Endoplasmic Reticulum Stress: The Key to Drug-Resistance of Multiple Myeloma?"

_cancers, 2022, doi:10.3390/cancers14215340_

Round 1

Reviewer 1 Report

Very comprehensive review of ERS in multiple myeloma in the preclinical setting. I would recommend a revision of english language and style as in some instances the tone is rather informal. In addition please see the following:

1. Need to correct/ revise: 

Line 43" resistance to therapies after one or more cycles of treatments" not cycles, your mean lines of treatment. 

Line 44-47 "In the up-to-date guidelines for MM [5], frontline drugs include bortezomib, cyclophosphamide, lenalidomide, melphalan and dexamethasone, together with the newly adapted monoclonal antibody Daratumumab. Despite deep remission for a period of time, the duration of response decreases with each line of therapy [6]. Patients who are clinically resistant to both bortezomib and first-line IMiDs have poorer outcome [7]

This paragraph needs rewording, for example you need to clarify that drug combinations are used in first line, doublet or triplet and not just list all possible agents used in first line. In addition don’t state “deep remission” as the depth of response is variable and can vary from primary refractory, to partial response to complete remission. So please rephrase the paragraph. 

Line 52: “deposit into blood and in”. Immunoglobulins cannot deposit into blood, they are secreted and circulate. Also

 “and in tissue, leading to organ dysfunction” they only lead to kidney dysfunction and cast nephropathy. Bone disease, hypercalcemia and anemia have other mechanisms. Please rephrase 

Line 208

“Due to the nature of MM cells, most backbone treatment strategies of MM are based 208 on proteasome inhibitors (PIs)” Need to discuss IMiDs in addition to proteasome inhibitors 

2. The review is very long, I recommend reducing the word count and also reducing the number of references as the list is also too long. 

3. Please consider expanding the discussion and perspectives with regards to the potential significance of all these data in the actual clinical/ patient setting. I

4. Are there any data regarding ERS and Immunomodulatory agents which are the group of drugs very widely used in MM and which affects ubiquitination. 

Reviewer 2 Report

                This is a comprehensive, extremely detailed review of recent treatments for refractory multiple myeloma (MM). Wang et al. place special emphasis on exploiting the endoplasmic reticulum stress response (ESR) to achieve a therapeutic response.  The review should be of interest to your readers. However, some modification is needed to enhance reader understanding.  Specific points include the following:

Although the review is generally well-written, there are numerous instances of incorrect English usage. For example, line 67 (repairment), Line 113, denutrition, line 159 arouses should be induces, line 573 “HSP90 inhibitors have been enrolled in clinical trials” etc. A thorough editing is needed before publication.

The authors cite a 2020 review (line 38) in which the incidence rates of MM vary from 0.1 to 5.3 per 100,00. Some brief explanation of this discrepancy should be noted.

Line 287 The authors very briefly report that “adherence of MM cells to tumor environment” may be involved in drug resistance. This is an intriguing observation and should be briefly expanded.

In my copy, Table 1 is misaligned.  The heading seems to span two lines. It is also unclear what “research information in MM” means. I would also suggest completing drug target for every agent, even though this may be redundant in some cases.

Author Response

Response to Reviewer 2 Comments

This is a comprehensive, extremely detailed review of recent treatments for refractory multiple myeloma (MM). Wang et al. place special emphasis on exploiting the endoplasmic reticulum stress response (ESR) to achieve a therapeutic response.  The review should be of interest to your readers. However, some modification is needed to enhance reader understanding.  Specific points include the following:

Response: We thank the reviewer for the valuable comments on our review article.

Point 1. Although the review is generally well-written, there are numerous instances of incorrect English usage. For example, line 67 (repairment), Line 113, denutrition, line 159 arouses should be induces, line 573 “HSP90 inhibitors have been enrolled in clinical trials” etc. A thorough editing is needed before publication.

Response: The English language has been revised according to the constructive suggestions. In marked-up revised manuscript, the sentences containing the word “arouses” has been omitted according to the comments by the other reviewers. As suggested, we have replaced “denutrition” with “nutrition deprivation” in line 129, “repairment” with “repair” in line 81 and “enrolled” with “investigated”, “tested” and “evaluated” in lines 514, 623, 676, 678 and 685. In addition, a thorough editing has been done to improve the English usage in revised manuscript.

Point 2. The authors cite a 2020 review (line 38) in which the incidence rates of MM vary from 0.1 to 5.3 per 100,00. Some brief explanation of this discrepancy should be noted.

Response: According to the valuable suggestion, we have added brief explanation of the discrepancy in marked-up revised manuscript, as “According to epidemiologic data from the International Agency for Research on Cancer, the current worldwide age-standardized incidence rates of MM vary from 0.54 to 5.3 per 100,000, which is associated with economic resources, patient education and quality of health care [2].” on page 2, lines 36-39.

Point 3. Line 287 The authors very briefly report that “adherence of MM cells to tumor environment” may be involved in drug resistance. This is an intriguing observation and should be briefly expanded.

Response: In marked-up revised manuscript we have expanded the description regarding cell adhesion-mediated drug resistance in MM accordingly, as “It has been well established that both cellular and non-cellular components of bone marrow microenvironment promote the development of drug resistance of MM [70,71]. Adherence of MM cell to bone marrow stromal cells (BMSCs) or fibronectin has been implicated in BZ and melphalan resistance via integrin-dependent upregulation of HSP70 [72], indicating a role of ERSR signaling in cell adhesion-mediated drug resistance in MM.”. Please see page 8, lines 322-327.

Point 4. In my copy, Table 1 is misaligned.  The heading seems to span two lines. It is also unclear what “research information in MM” means. I would also suggest completing drug target for every agent, even though this may be redundant in some cases.

Response: As suggested, we have modified the content and format of Table 1, and changed “research information in MM” to “research status in MM”. Please see revised Table 1 in Section 5 “Emerging strategies to overcome MM drug resistance by targeting ERS” on page 14.

References:

[2] Ludwig, H.; Novis Durie, S.; Meckl, A.; Hinke, A.; Durie, B. Multiple Myeloma Incidence and Mortality Around the Globe; Interrelations Between Health Access and Quality, Economic Resources, and Patient Empowerment. Oncologist 2020, 25, e1406-e1413, doi:10.1634/theoncologist.2020-0141.

[70] Hideshima, T.; Mitsiades, C.; Tonon, G.; Richardson, P.G.; Anderson, K.C. Understanding multiple myeloma pathogenesis in the bone marrow to identify new therapeutic targets. Nat Rev Cancer 2007, 7, 585-598, doi:10.1038/nrc2189.

[71] Ria, R.; Vacca, A. Bone Marrow Stromal Cells-Induced Drug Resistance in Multiple Myeloma. Int J Mol Sci 2020, 21, doi:10.3390/ijms21020613.

[72] Nimmanapalli, R.; Gerbino, E.; Dalton, W.S.; Gandhi, V.; Alsina, M. HSP70 inhibition reverses cell adhesion mediated and acquired drug resistance in multiple myeloma. Br J Haematol 2008, 142, 551-561, doi:10.1111/j.1365-2141.2008.07217.x.

Reviewer 3 Report

Interesting and comprehensive study regarding the mechanisms of ERSR signaling in MM cells resistant to conventional chemotherapy. The aim was to find new targets for salvage therapy. However, too many details are reported in the paper not always useful for a targeted therapy. For increasing the clinical relevance of the study the Authors should make an effort to report  only the data regarding the signaling mechanisms which have been tested by targeted therapy on cell lines and clinical study.

Author Response

Response to Reviewer 3 Comments

Interesting and comprehensive study regarding the mechanisms of ERSR signaling in MM cells resistant to conventional chemotherapy. The aim was to find new targets for salvage therapy. However, too many details are reported in the paper not always useful for a targeted therapy. For increasing the clinical relevance of the study the Authors should make an effort to report only the data regarding the signaling mechanisms which have been tested by targeted therapy on cell lines and clinical study.

Response: We thank the reviewer for the constructive comments to improve the quality of our review article. As suggested, major revision has been made to our manuscript. In marked-up revised manuscript, we have omitted details which are not useful for a targeted therapy from the revised manuscript, and expanded the discussion and perspectives on the potential clinical significance of the signaling mechanisms relevant to targeted therapy. Moreover, we have included the data regarding ERSR signaling and Immunomodulatory agents, as well as the description of ERSR signaling and immunotherapy, and added related literature.

Round 2

Reviewer 1 Report

I am happy to accept the paper in its present form after the revisions. 

Reviewer 3 Report

The revised paper clarifies better the possible role of modulating ERSR signaling for overcoming drug resistance in MM patients and the application of basic research in designing new targeted therapies.